# Population-Based External Validation of the EASIX Scores to Predict CAR T-Cell-Related Toxicities

**DOI:** 10.3390/cancers15225443

**Published:** 2023-11-16

**Authors:** Janneke W. de Boer, Kylie Keijzer, Elise R. A. Pennings, Jaap A. van Doesum, Anne M. Spanjaart, Margot Jak, Pim G. N. J. Mutsaers, Suzanne van Dorp, Joost S. P. Vermaat, Marjolein W. M. van der Poel, Lisanne V. van Dijk, Marie José Kersten, Anne G. H. Niezink, Tom van Meerten

**Affiliations:** 1Department of Hematology, University of Groningen, University Medical Center Groningen, 9713 GZ Groningen, The Netherlands; j.w.de.boer@umcg.nl (J.W.d.B.); k.keijzer@umcg.nl (K.K.); j.a.van.doesum@umcg.nl (J.A.v.D.); 2Department of Radiation Oncology, University of Groningen, University Medical Center Groningen, 9713 GZ Groningen, The Netherlands; l.v.van.dijk@umcg.nl (L.V.v.D.); a.g.h.niezink@umcg.nl (A.G.H.N.); 3Department of Hematology, Amsterdam UMC Location University of Amsterdam, 1007 MB Amsterdam, The Netherlandsm.j.kersten@amsterdamumc.nl (M.J.K.); 4Cancer Center Amsterdam, 1105 AZ Amsterdam, The Netherlands; 5LYMMCARE (Lymphoma and Myeloma Center Amsterdam), 1105 AZ Amsterdam, The Netherlands; 6Erasmus School of Health Policy and Management, Erasmus University Rotterdam, 3062 PA Rotterdam, The Netherlands; 7Department of Hematology, University Medical Center Utrecht, 3584 CX Utrecht, The Netherlands; m.jak@umcutrecht.nl; 8Department of Hematology, Erasmus MC Cancer Institute, University Medical Center Rotterdam, 3015 GD Rotterdam, The Netherlands; p.mutsaers@erasmusmc.nl; 9Department of Hematology, Radboud University Medical Center, 6500 HB Nijmegen, The Netherlands; suzanne.vandorp@radboudumc.nl; 10Department of Hematology, Leiden University Medical Center, 2333 ZA Leiden, The Netherlands; j.s.p.vermaat@lumc.nl; 11Department of Internal Medicine, Division of Hematology, GROW School for Oncology and Developmental Biology, Maastricht University Medical Center, 6229 HX Maastricht, The Netherlands; marjolein.vander.poel@mumc.nl

**Keywords:** CAR T-cell therapy, toxicity, EASIX, LBCL

## Abstract

**Simple Summary:**

CAR T-cell therapy became standard of care for patients with relapsed or refractory large B-cell lymphoma. However, their administration can be accompanied by toxicities, such as cytokine release syndrome and immune effector cell-associated neurotoxicity syndrome. It is important to identify patients at risk for these toxicities in order to start an early intervention in high-risk patients and guide outpatient CAR T-cell treatment. As a consequence, several easy-to-use risk scores including the EASIX and its derivatives were developed. However, in the available studies, disparities existed among the used endpoints and cutoff values, hampering the utility of these tools in practice. This study aims to validate these EASIX scores in a population-based cohort. This can be used to select the best predictive model and to further guide optimization of the proposed risk scores.

**Abstract:**

Cytokine release syndrome (CRS) and immune effector cell-associated neurotoxicity syndrome (ICANS) can hamper the clinical benefit of CAR T-cell therapy in patients with relapsed/refractory large B-cell lymphoma (r/r LBCL). To assess the risk of CRS and ICANS, the endothelial activation and stress index (EASIX), the modified EASIX (m-EASIX), simplified EASIX (s-EASIX), and EASIX with CRP/ferritin (EASIX-F(C)) were proposed. This study validates these scores in a consecutive population-based cohort. Patients with r/r LBCL treated with axicabtagene ciloleucel were included (*n* = 154). EASIX scores were calculated at baseline, before lymphodepletion (pre-LD) and at CAR T-cell infusion. The EASIX and the s-EASIX at pre-LD were significantly associated with ICANS grade ≥ 2 (both *p* = 0.04), and the EASIX approached statistical significance at infusion (*p* = 0.05). However, the predictive performance was moderate, with area under the curves of 0.61–0.62. Validation of the EASIX-FC revealed that patients in the intermediate risk group had an increased risk of ICANS grade ≥ 2 compared to low-risk patients. No significant associations between EASIX scores and CRS/ICANS grade ≥ 3 were found. The (m-/s-) EASIX can be used to assess the risk of ICANS grade ≥ 2 in patients treated with CAR T-cell therapy. However, due to the moderate performance of the scores, further optimization needs to be performed before broad implementation as a clinical tool, directing early intervention and guiding outpatient CAR T-cell treatment.

## 1. Introduction

Chimeric antigen receptor (CAR) T-cell therapy is an immunotherapy in which autologous T-cells are genetically engineered to recognize a particular antigen. Axicabtagene ciloleucel (axi-cel), a CD19-directed CAR T-cell product, was shown to be highly effective in patients with relapsed/refractory large B-cell lymphoma (r/r LBCL) compared to standard second- or third-line immunochemotherapy [1,2]. However, patients may experience severe acute toxicities, hampering their clinical benefit [3,4]. These toxicities consist of the immune-related adverse events cytokine release syndrome (CRS) and immune effector cell-associated neurotoxicity syndrome (ICANS) [5].

Prediction of patients at risk for severe CRS and/or ICANS is essential to enable early intervention with tocilizumab and/or steroids in this specific subgroup and thereby limit the severity, as well as the need and costs for intensive care admission. In addition, adequate prediction could support the implementation of CAR T-cell therapy on an outpatient basis. Early intervention, whether combined with prophylactic steroid administration or used as a standalone intervention, was already confirmed to reduce rates of CRS and ICANS, and lower the cumulative dose of steroids administered, reducing the total dose of immunosuppression and therefore potentially reducing the risk of infections [6,7].

Recent studies reported different biomarkers associated with the occurrence of CRS and/or ICANS, such as lactate dehydrogenase (LDH), as marker for tumor burden, the inflammatory markers C-reactive protein (CRP) and ferritin, and platelet count as an indicator for hematopoietic reserve [8,9,10]. In addition, elevated levels of circulating cytokines interleukin (IL)-6, IFN-y, IL-10, IL-15, and monocyte chemoattractant protein (MCP)-1 are associated with severity of CAR T-cell-related toxicity [10,11,12,13]. These are all known surrogate markers for endothelial dysfunction in capillary leak syndromes, such as sepsis and acute respiratory distress syndrome [11]. Therefore, a pivotal role of endothelial dysfunction in CRS and/or ICANS was suggested [9,14,15].

The endothelial activation and stress index (EASIX) and its derivatives, the modified EASIX (m-EASIX), simplified EASIX (s-EASIX), and the risk stratification algorithm EASIX-ferritin/CRP (EASIX-F(C)) are proposed as easy-to-use clinical tools including markers for endothelial dysfunction to assess the risk of severe CRS and ICANS in patients with r/r LBCL [8,16,17,18]. However, in the available studies, disparities existed among the endpoint definitions, as severe CRS or ICANS were defined either as grade ≥ 3 or as grade ≥ 2, or a combination endpoint was used of CRS and/or ICANS grade ≥ 3. Moreover, no validation of cut-off points was performed and no consensus was reached on the best predictive model, creating difficulty in implementation of the EASIX scores for identifying patients at risk in daily practice.

In this study, we aim to externally validate these scores in a consecutive population-based cohort to establish the utility of these scores to predict severe toxicity in the real world. Adequate prediction will enable early risk-adapted prevention strategies and/or selection of patients for CAR T-cell treatment on an outpatient basis.

## 2. Materials and Methods

### 2.1. Patient Population

In this population-based cohort study, all patients with r/r LBCL after ≥2 lines of systemic therapy who received axi-cel as standard of care or in an early access program between June 2019 and May 2022 in the Netherlands were included. For patients who received axi-cel as standard of care, eligibility for CAR T-cell therapy was approved by the Dutch CAR-T tumorboard according to the criteria described previously [19]. Patients received lymphodepleting chemotherapy with cyclophosphamide (500 mg/m^2^) and fludarabine (30 mg/m^2^) for three consecutive days (Day-5, -4, and -3) followed by a single infusion of CAR T-cells (Day 0).

Patient data and the laboratory values lactate dehydrogenase (LDH), creatinine, platelets, C-reactive protein (CRP), and ferritin were collected from tumorboard referral forms and medical records at all Dutch CAR T-cell treating centers. The study was conducted in accordance with the Declaration of Helsinki and approved by the Ethics Committee of the Academic Medical Center (NL76835.018.21). Informed consent was obtained from all patients according to the national guidelines.

### 2.2. CRS/ICANS Grading and Endpoints

CRS and ICANS were prospectively graded according to the American Society for Transplantation and Cellular Therapy (ASTCT) grading system [20]. Tocilizumab and steroids were administered based on the CARTOX treatment protocol or on the early intervention protocol [2,6,7]. To perform an external validation, endpoints were defined as reported previously: namely CRS grade ≥ 2 and ICANS grade ≥ 2, CRS grade ≥ 3 and ICANS grade ≥ 3, and a combination endpoint of CRS and/or ICANS grade ≥ 3 [8,16,17,18].

### 2.3. Statistical Analysis

The EASIX and its derivatives were calculated as per original reports: the EASIX as [(creatinine × LDH)/platelets], the m-EASIX as [(CRP × LDH)/platelets], and the s-EASIX as [CRP/platelets]. All scores were determined at three different timepoints: at screening or apheresis (baseline), before start lymphodepleting regimen (pre-LD, day-15 until day-5), and at infusion (day 1 ± 1 day) [8,16,17,18]. The Mann–Whitney U test was used to compare EASIX scores between patients having CRS/ICANS grade < 2 and patients having CRS/ICANS grade ≥ 2. In addition, scores were also compared between patients having CRS/ICANS grade < 3 and patients having CRS/ICANS grade ≥ 3. Missing laboratory parameters were imputed using predictive mean matching with ten imputation datasets. To reduce skewness, a log2 transformation was applied to all the EASIX formats and laboratory parameters, as published in the original reports [16,17,18]. Univariable logistic regression was performed to define the association of the EASIX and its derivatives and other laboratory parameters with the described endpoints. Receiver operating characteristics (ROC) curve analyses, including area under the curve (AUC) calculations, were used to investigate the predictive performance of the EASIX and its derivatives for the different endpoints. Furthermore, external validation of the risk stratification algorithm presented by Greenbaum et al. was performed to predict development of CRS grade ≥ 2 and ICANS grade ≥ 2 using Fine–Gray regression that accounted for death as a competing event [8]. Pooled estimates of analyses on the imputed datasets were reported by using Rubin’s rules. A *p*-value ≤ 0.05 was considered statistically significant. All statistical analyses were performed using R version 4.2.1 (R foundation for statistical computing, Vienna, Austria).

## 3. Results

The external validation performed in this study was based on the four available studies describing the EASIX and/or a derivative of the EASIX [8,16,17,18]. Detailed information regarding these studies is presented in Table 1.

### 3.1. Patient and Treatment Characteristics

A population-based cohort consisting of 154 r/r LBCL patients who received axi-cel was included in this study. The median age was 60 years (range 18–84) and the majority of the patients was male (65%). Roughly half of the patients were diagnosed with DLBCL (51%), followed by tFL (33%), HGBCL (13%), and PMBCL (3%). At screening, the ECOG performance score was 0–1 in 90% of the patients. Disease stage III or IV was observed in 80% of patients and 65% had extranodal localizations. The International Prognostic Index (IPI) was intermediate-high in 28% and high in 8% of patients.

In the first 30 days after infusion, 91% of patients experienced any grade of CRS, whereof 45% experienced a grade 2 or higher. Tocilizumab was given to 70% of the patients. Any grade of ICANS was observed in 60% of patients, of which 41% had a grade 2 or higher, and 62% of patients received steroids. A small fraction of patients was ICU admitted (14%) due to CRS and/or ICANS development in 17 of the 21 cases. The cause for admission of one patient was unknown, whereas for the other patients, the reason for admission included either sepsis, bradycardia, or a gastrointestinal bleeding. For two patients, the main cause of death was due to ICANS. Detailed patient and treatment characteristics are provided in Table 2.

Regarding the collection of baseline laboratory values, median time from screening to collection was 6 days and the median time from baseline to pre-LD collection was 28 days (IQR [22–32]).

### 3.2. EASIX/m-EASIX/s-EASIX Distributions

Pre-LD EASIX, m-EASIX, and s-EASIX were grouped according to CRS and ICANS grades 2 or higher, and are depicted in Figure 1 (*n* = 152 for EASIX and s-EASIX, *n* = 141 for m-EASIX). No significant changes in (m-/s-) EASIX were observed between patients experiencing CRS grade < 2 versus grade ≥ 2. The EASIX scores were significantly higher for patients with ICANS grade ≥ 2 compared to patients with grade < 2 (*p* < 0.01). This was also observed for the m-EASIX (*p* = 0.03) and s-EASIX scores (*p* = 0.02). No significant differences in the mean of the EASIX scores were observed for CRS grade < 3 versus CRS grade ≥ 3 and ICANS grade < 3 versus ICANS grade ≥ 3. Summary statistics of the EASIX scores at baseline, pre-LD and infusion, and additionally grouped according to CRS and/or ICANS grade ≥ 3 are presented in Appendix A. 

**Table 1 cancers-15-05443-t001:** Literature overview.

Author	Year	Patients	Timepoint	Outcome	Used Variable	Variable Form	Statistical Method	Association Descriptives	Performer Descriptives	Conclusion
Greenbaum et al. [8]	2021	r/r LBCL patients treated with axicel (*n* = 171).	Pre-LD	CRS ≥ grade 2	EASIX-F	EASIX: >4.6 ferritin: >321 ng/mL	Training model: Fine and Gray regression analyses Validation: bootstrapping 3000 resampled data	EASIX: HR 2.4, *p* < 0.001 for >UQ	EASIX-F identified 3 risk groups with cumulative incidence of 74% (*p* < 0.001), 51% (*p* = 0.04) and 23% (reference)	EASIX combined with ferritin could discriminate three different risk groups for CRS grade 2–4.
				ICANS grade ≥ 2	EASIX-FC	EASIX: >2.1 ferritin: >1583 ng/mL CRP: >21 mg/L		EASIX: HR 2.2, *p* < 0.001 for >median	EASIX-FC identified 3 risk groups with cumulative incidence of 74% (*p* < 0.001), 51% (*p* = 0.025) and 29% (reference)	EASIX combined with CRP and ferritin could significantly discriminate three different risk groups for ICANS grade 2–4.
Pennisi et al. [16]	2021	B-ALL treated with 1928z CAR T cells and r/r LBCL patients treated with axicel and tisacel (*n* = 118).	Pre-LD D − 1 D + 1 D + 3	CRS ≥ grade 3	EASIX (log2)	Continuous	Logistic regression AUC	Pre-LD: OR 1.34, s D − 1: OR 1.51, s D + 1: OR 1.56, s D + 3: OR 1.89, s	Pre-LD: AUC 0.77 D − 1: AUC 0.72 D + 1: AUC 0.72 D + 3: AUC 0.80	EASIX, m-EASIX and s-EASIX were significantly associated with the occurrence of severe CRS on multiple time points. All three formulas were able to predict severe CRS well.
					m-EASIX (log2)			Pre-LD: OR 1.32, s D − 1: OR 1.26, s D + 1: OR 1.31, s D +3: 1.56, s	Pre-LD: AUC 0.80 D − 1: AUC 0.73 D + 1: AUC 075 D + 3: AUC 0.73	
					s-EASIX (log2)			Pre-LD: OR 1.49, s D − 1: OR 1.6, s D + 1: OR 1.65, s D + 3: OR 1.92, s	Pre-LD: AUC 0.82 D − 1: AUC 0.75 D + 1: AUC 0.76 D + 3: AUC 0.81	
				ICANS ≥ grade 3	EASIX (log2)	Continuous	Logistic regressionAUC	Pre-LD: OR 1.11, ns D − 1: OR 1.2, ns D + 1: OR 1.36, s D + 3 OR 1.5, s	D + 1: AUC 0.61 D + 3: AUC 0.68	EASIX, m-EASIX and s-EASIX on day +1 and +3 were significantly associated with the occurrence of severe ICANS. The predictive power of these three formulas on day +1 and +3 was moderate.
					m-EASIX (log2)			Pre-LD: OR 1.1, ns D − 1: OR 1.12, ns D + 1: OR 1.2, s D + 3: OR 1.36, s	D + 1: AUC 0.67 D + 3: AUC 0.73	
					s-EASIX (log2)			Pre-LD: OR 1.25, ns D − 1: OR 1.33, ns D + 1: OR 1.46, s D + 3: OR 1.55, s	D + 1: AUC 0.66 D + 3: AUC 0.68	
Korell et al. [17]	2022	Training cohort: r/r LBCL patients treated with axicel (*n* = 93). Validation cohort: r/r LBCL/MCL/ ALL/FL/CLL patients treated with axi-cel/tisa-cel or HD-CAR-1 (*n* = 121).	Pre-LD	CRS/ICANS ≥ grade 3	EASIX (log2)	Continuous Cut-off point 4.67	Multivariate logistic regression Validation cohort: AUC, Brier scores	Continuous: OR 1.72 *p* = 0.001 ^†^ Cut-off > 4.67: OR 4.32, *p* = 0.006 ^†^	Continuous: AUC 0.81	EASIX, s-EASIX and m-EASIX pre-LD were significantly associated with CRS or ICANS grade ≥ 3. All three formulas could predict the occurrence of toxicity and out-performed the reference model in multivariate analysis.
					m-EASIX (log2)	Continuous		OR 1.22 *p* = 0.015 ^†^	AUC 0.74	
					s-EASIX (log2)			OR 1.63, *p* = 0.004 ^†^	AUC 0.79	
Acosta-Medina et al. [18]	2023	r/r LBCL patients treated with axicel (*n* = 84).	Pre-LD D0	ICANS ≥ grade 3	EASIX	Continuous	Univariable logistic regressionAUC	Continuous: Pre-LD: OR 1.14, *p* = 0.047 D0: OR 1.19, *p* = 0.008	Continuous: Pre-LD: 0.57 D0: 0.62	EASIX and m-EASIX were associated with increased risk of ICANS G3–4 at lymphodepletion, but were further optimized when calculated from laboratory values at infusion. Only m-EASIX at infusion was able to categorically predict high-risk patients.
					m-EASIX	Continuous, Cut-off point 4		Continuous: Pre-LD: OR 1.007, *p* = 0.205 D0: OR 1.007, *p* = 0.086 Cut-off ≥ 4: D0: OR 4.086, *p* = 0.034	Continuous: D0: 0.72	

EASIX = endothelial activation and stress index; m-EASIX = modified EASIX score; s-EASIX = simplified EASIX score; EASIX-F = EASIX + ferritin; EASIX-FC = EASIX + ferritin + CRP; pre-LD = pre-lymphodepleting chemotherapy (day-15 until day-5); D − 1 = one day before infusion; D + 1 = one day after infusion; D + 3 = three days after infusion; D0 = day of infusion; AUC = area under the curve; ^†^ corrected for age, gender, diagnosis, and disease status.

**Table 2 cancers-15-05443-t002:** Patient and treatment characteristics.

	Total (*n* = 154)
**Age**, median (range)	60 (18–84)
**Gender, male, *n* %**	101 (65.6)
**Diagnosis**, *n* %	
DLBCL	79 (51.3)
tFL	50 (32.5)
HGBCL DH/TH	14 (9.1)
HGBCL NOS	6 (3.9)
PMBCL	5 (3.2)
**ECOG**, *n* %	
0–1	138 (89.6)
2–4	11 (7.1)
Missing, *n* %	5 (3.3)
**Disease stage** ^a^**, *n* %**	
Stage I–II	34 (22.1)
Stage III–IV	120 (77.9)
**Bulky disease** ^a^, *n* %	51 (33.1)
Missing, *n* %	3 (2.0)
**Nr. of extranodal sites** ^a^, *n* %	
0	52 (33.8)
1	55 (35.7)
≥2	45 (29.2)
Missing, *n* %	2 (1.3)
**LDH at screening**, median (IQR)	269 (215–446)
Missing, *n* %	15 (9.7)
**LDH at lymphodepletion**, median (IQR)	238 (195–329)
Missing, n%	2 (1.3)
**IPI** ^a^, *n* %	
Low	32 (20.8)
Low-intermediate	45 (29.2)
Intermediate-high	43 (27.9)
High	13 (8.4)
Missing, *n* %	21 (14)
**Patients refractory to first-line treatment** ^b^, *n* %	94 (61.0)
**Patients refractory to second-line treatment** ^b^, *n* %	114 (74.0)
Missing, *n* %	12 (7.8)
**Previous lines of therapy**, median (range)	2 (2–10)
**Previous stem cell transplant**, *n* %	45 (29.2)
Allogenic	3 (1.9)
Autologous	45 (29.2)
**Bridging therapy**, *n* %	
No bridging	32 (20.8)
Radiotherapy	37 (24.0)
Systemic therapy	34 (22.1)
Steroids	19 (12.3)
Combination	32 (20.8)
**CRS grade**, *n* %	
No CRS	14 (9.1)
1	71 (46.1)
2	61 (39.6)
3	7 (4.5)
4	1 (0.6)
**ICANS grade**, *n* %	
No ICANS	61 (39.6)
1	30 (19.5)
2	31 (20.1)
3	28 (18.2)
4	4 (2.6)

^a^ determined at screening; ^b^ primary refractory was defined as no complete response to treatment or relapse within 12 months after treatment. CRS = cytokine release syndrome; DH/TH = double hit or triple hit; DLBCL = diffuse large B cell lymphoma; HGBCL = high-grade B-cell lymphoma; ICANS = immune effector cell-associated neurotoxicity syndrome; IPI = International Prognostic Index; IQR = interquartile range; LDH = lactate dehydrogenase; NOS = not otherwise specified; PMBCL = primary mediastinal large B-cell lymphoma; and tFL = transformed follicular lymphoma.

### 3.3. Univariable Associations with CRS and ICANS Development

The EASIX at pre-LD was significantly associated with ICANS grade ≥ 2 (OR 1.31 CI [1.02–1.68]; *p* = 0.04) and at infusion approached statistical significance (OR 1.28 CI [1.00–1.62]; *p* = 0.05) (Figure 2).

In addition, the s-EASIX at pre-LD was significantly associated with ICANS grade ≥ 2 (OR 1.33 CI [1.02–1.73]; *p* = 0.04). Borderline associations were retrieved with the m-EASIX at pre-LD (OR 1.11 CI [0.99–1.25]; *p* = 0.06), and the s-EASIX at baseline (OR 1.29 CI [0.98–1.70]; and *p* = 0.07) and infusion (OR 1.28 CI [0.99–1.67]; *p* = 0.06). No associations between EASIX scores and CRS grade ≥ 2 were found. Additionally, we did not find any associations for CRS and/or ICANS grade ≥ 3 (Table 3, Appendix A) [16,17,18].

Laboratory parameters that were associated with CRS grade ≥ 2 were LDH at pre-LD and infusion (OR 2.00 CI [1.19–1.35]; *p* = 0.01 and OR 1.63 CI [1.00–2.64]; *p* = 0.05); for ICANS grade ≥ 2 associations were found with platelets at baseline and infusion (OR 0.63 CI [0.44–0.91]; *p* = 0.02 and OR 0.73 CI [0.53–1.00]; *p* = 0.05), and ferritin at infusion (OR 1.29 CI [1.00–1.67]; *p* = 0.05; Appendix A). Clinical factors associated with toxicity were male gender for CRS grade ≥ 3, ICANS grade ≥ 3, and CRS/ICANS grade ≥ 3 (OR 0.42 CI [0.21–0.83]; *p* = 0.01, OR 0.44 CI [0.20–0.96]; and *p* = 0.04 and 0.41 CI [0.20–0.88]; *p* = 0.02), ECOG performance status > 1 for CRS grade ≥ 3 (OR 6.66 CI [1.08–41.05]; *p* = 0.04), and IPI-score > 1 for ICANS grade ≥ 2 (OR 3.15 CI [1.03–9.67]; *p* = 0.05). Detailed information can be found in Appendix A.

### 3.4. ROC Curve Analysis

ROC curve analyses were performed where the best performance was achieved with EASIX at pre-LD for predicting ICANS grade ≥ 2 (AUC = 0.62; Figure 3). Similar performances for ICANS grade ≥ 2 were found with s-EASIX at pre-LD (AUC = 0.61), s-EASIX at infusion (AUC = 0.61), and EASIX at infusion (AUC = 0.61). Evaluations of the performance of (m-/s-) EASIX at all timepoints to predict CRS and ICANS grade ≥ 2 are shown in Figure 3.

### 3.5. EASIX Risk-Stratification

Evaluation of the risk-stratification algorithm proposed by Greenbaum et al. using pre-LD EASIX, ferritin, and CRP (EASIX-FC) to predict ICANS grade ≥ 2 development showed a significant association between the intermediate risk group and ICANS grade ≥ 2, compared to the low-risk group (HR 2.04 CI [1.26–3.32]; *p* < 0.01; Table 4) [8]. The cumulative incidence of the intermediate-risk group including a high EASIX was leading (86%), indicating low ferritin levels and a high EASIX being discriminative of ICANS grade ≥ 2 development. However, no significant difference was found between the high and low-risk groups (*p* = 0.16), which may be influenced by a cumulative incidence of 57% in the high-risk group.

The risk stratification algorithm using pre-LD EASIX and ferritin to predict CRS grade ≥ 2 showed no significant associations between the different risk groups and CRS development. Additionally, cumulative incidences of the intermediate- and high-risk groups were lower (32%) and comparable (47%) to the low-risk group (43%), indicating no increased risk of CRS development.

### 3.6. EASIX Cutoff

In an additional analysis, we investigated the proposed threshold values with the corresponding endpoints. We could only find a significant association between the EASIX cutoff > 2.1 and ICANS ≥ grade 2, proposed by Greenbaum et al. prior to development of the risk stratification score (OR 3.24 CI [1.58–6.68]; *p* < 0.01) [8]. The corresponding predictive performance was moderate (AUC = 0.62). Detailed results can be found in Appendix A.

## 4. Discussion

The EASIX and its derivatives were reported as easy-to-use clinical tools to predict patients at risk for severe CRS and ICANS after CD19-directed CAR T-cell therapy in several studies [8,16,17,18]. However, the implementation of these scores in routine clinical practice is difficult, as there is no consensus on the best predictive model, and different cutoff points and endpoints are used among the performed studies. In addition, if this clinical tool is used for not only directing early immunosuppressive intervention for high-risk patients but also guiding outpatient CAR T-cell treatment, adequate predictive performance of CRS/ICANS grade ≥ 2 in addition to grade ≥ 3 is essential, as these patients already require supportive care.

This study is the first to report an external validation of all proposed EASIX scores in a population-based cohort. We showed an association of the EASIX, s-EASIX, and a trend towards significance for the m-EASIX calculated at pre-LD with ICANS grade ≥ 2, although the predictive performance of these scores is only moderate. In addition, the EASIX calculated at infusion approached statistical significance for ICANS grade ≥ 2; but again, the predictive performance was moderate. No associations could be observed between the EASIX and its derivatives measured at baseline and ICANS ≥ grade 2. In line with Pennisi et al., but in contrast to Acosta-medina et al., a significant association between the (m/s-) EASIX at pre-LD and ICANS grade ≥ 3 was not observed [16,18].

Regarding the risk stratification strategies proposed by Greenbaum et al., the risk score could not be validated for CRS grade ≥ 2, but for ICANS grade ≥ 2 the intermediate risk group (low ferritin/high EASIX) was predictive for a higher risk [8]. However, when the intermediate- and high-risk groups were combined in an additional analysis, cumulative incidence dropped to 60% in this group compared to the low-risk group (HR 1.94 CI [1.14–3.32]; *p* = 0.02), suggesting a lack of discriminative predictive ability although statistically significant.

In contrast to the existing literature, an association of the EASIX or its derivatives was not observed for the endpoint CRS ≥ grade 3 or the combined CRS/ICANS ≥ grade 3 endpoint [16,17]. Our cohort comprises a lower percentage of patients with CRS ≥ grade 3 (5%) and ICANS ≥ grade 3 (21%) compared to the already published cohorts, despite the fact that it is relatively large and includes solely real-world data. This can be explained by several reasons. First, because our cohort has a later inclusion period, more aggressive toxicity management strategies were used, following the publication in 2021 of the results of the ZUMA-1 cohort 4 and 6, describing a reduced incidence of grade ≥ 3 CRS and ICANS without diminished efficacy with the early administration of corticosteroids and tocilizumab [6,7]. This emphasizes the necessity of adapting risk scores to new insights in the management of CAR T-cell therapy-related toxicities.

Second, eligibility of patients for CAR T-cell therapy in the Netherlands is assessed and approved by the Dutch CAR-T tumorboard. This expert-directed patient selection could have resulted in a more strict assessment of eligibility for CAR T-cell therapy and led to the exclusion of patients with very rapidly progressive diseases, who are prone to the occurrence of (high-grade) CAR T-cell therapy-related toxicity.

In addition to the variety of endpoints used across studies, there was a substantial degree of heterogeneity regarding the format of the EASIX scores, employed statistical methods to determine predictive performance, patient selection, and CAR T-cell product, as described in Table 1 [8,16,17,18]. Notably, variations were also evident in the cutoff values in these studies, hampering the utilization of this clinical tool in toxicity management. Only the threshold value of Greenbaum et al. could be validated, but predictive performance remained poor [8].

Originally, the EASIX was developed as a predictor for endothelial complications, such as graft-vs-host disease and sinusoidal obstruction syndrome, and mortality after allogenic stem cell transplantation [21,22,23,24,25]. In the context of CAR T-cell therapy, endothelial activation seems to play a pivotal role in the development and exacerbation of CRS and more profound ICANS [9,14,15]. In ICANS, endothelial activation causes the eruption of the blood–brain barrier and increased vascular permeability, which are both essential steps in the pathogenesis of this condition [5,9,14,26]. Moreover, a higher angiopoietin-2/angiopoietin-1 balance, indicating endothelial activation, at pre-LD was identified in patients subsequently developing ICANS, suggesting endothelial dysfunction prior to CAR T-cell therapy could influence the occurrence of toxicity [9]. In line with this, we could identify an association with platelets related to endothelial damage and complement activation and ferritin, an inflammatory marker, at pre-LD with ICANS ≥ grade 2. However, the exact attribution of pre-infusion endothelial activation to the incidence and severity of ICANS is unknown.

In contrast to ICANS, we could only identify an association of CRS grade ≥ 2 with the laboratory value LDH, a marker for tumor burden, indicating that especially a high tumor volume may trigger the occurrence of CRS. Higher metabolic tumor volume is indeed associated with the occurrence of CRS, but not with ICANS [27,28,29,30]. Therefore, adding the metabolic tumor volume to risk scores might enhance predictive performance for CRS.

A limitation of this study is that, although the presented analysis is performed in a population-based cohort, it is restricted to patients with r/r LBCL treated with axi-cel. Therefore, the application of these results to other diseases and CAR-T products is limited.

## 5. Conclusions

In conclusion, the EASIX, m-EASIX, s-EASIX, and EASIX-FC might be used to assess the risk of ICANS grade ≥ 2 in patients with r/r LBCL treated with CD19-directed CAR T-cell therapy. However, predictive performance is moderate and further optimization needs to be performed before broad implementation as a clinical tool in the current treatment landscape.

## Figures and Tables

**Figure 1 cancers-15-05443-f001:**
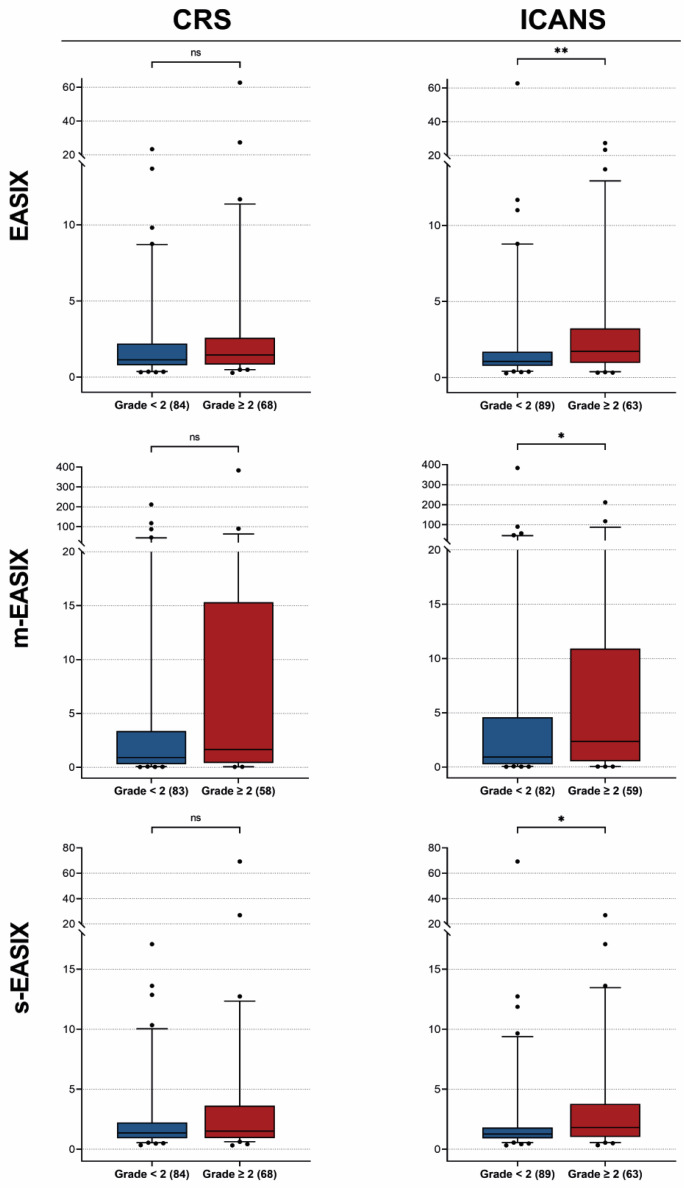
Distributions of EASIX/m-EASIX/s-EASIX scores at pre-LD across CRS and ICANS subgroups. Boxplots representing the median and interquartile range of (m-/s-) EASIX scores at pre-LD classified in the different CRS and ICANS subgroups: patients experiencing toxicity grade < 2 (blue) vs. patients experiencing toxicity grade ≥ 2 (red). Number of patients per subgroup are reported in brackets. ns = non-signicant, *p*-value > 0.05; * = *p*-value < 0.05; ** = *p*-value ≤ 0.01.

**Figure 2 cancers-15-05443-f002:**
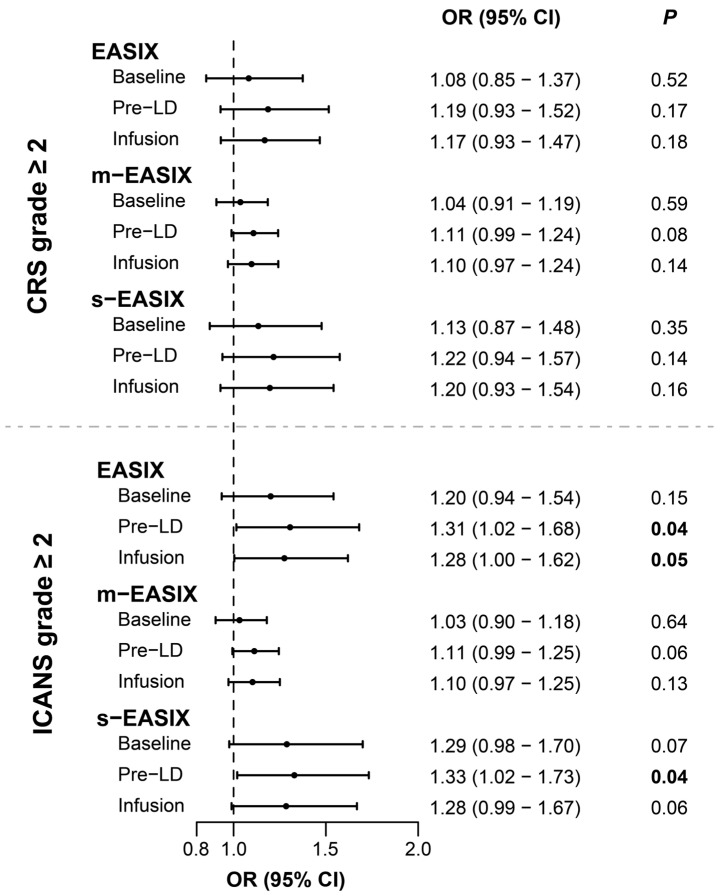
Associations of EASIX/m-EASIX/s-EASIX with CRS and ICANS grades ≥ 2. Odd ratios (ORs) with confidence intervals retrieved for the associations of (m-/s-) EASIX scores at baseline, pre-LD and infusion, with CRS (top) and ICANS (bottom) are expressed in a forest plot. ORs and *p*-values are additionally reported.

**Figure 3 cancers-15-05443-f003:**
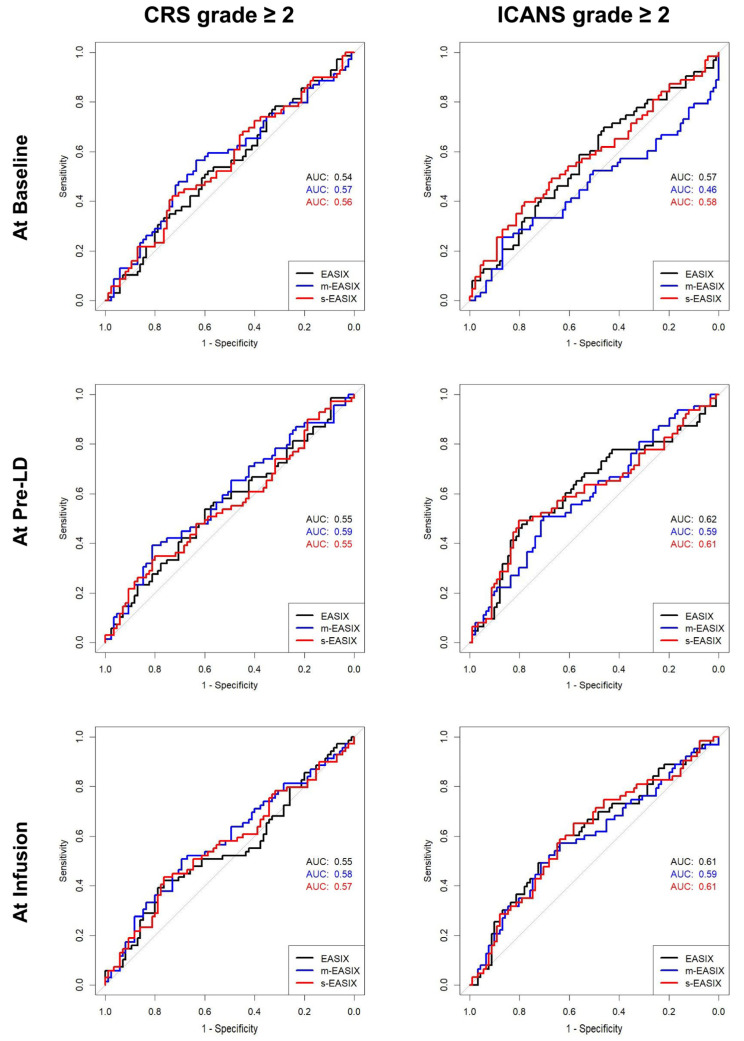
Prediction of CRS and ICANS grades ≥ 2 using EASIX/m-EASIX/s-EASIX. ROC curves with corresponding AUCs display the predictive performance of the (m-/s-) EASIX scores at baseline, pre-LD, and infusion.

**Table 3 cancers-15-05443-t003:** Overview of previous published results and our external validation for the EASIX-scores at pre-LD and different endpoints.

	CRS ≥ Grade 2		ICANS ≥ Grade 2		CRS ≥ Grade 3		ICANS ≥ Grade 3		CRS/ICANS ≥ Grade 3	
	Published ^‡^	Our Cohort	Published ^‡^	Our Cohort	Published	Our Cohort	Published	Our Cohort	Published ^+^	Our Cohort
EASIX ^†^	NR	0.17	NR	**0.04**	s	0.81	ns/**0.05**	0.45	0	0.71
m-EASIX ^†^	NR	0.08	NR	0.06	s	0.75	ns/0.21	0.59	**0.02**	0.99
s-EASIX ^†^	NR	0.14	NR	**0.04**	s	0.77	ns	0.58	0	0.87

NR = not reported/performed; s = significant association, no *p*-value reported; ns = non-significant association, no *p*-value reported; ^†^ log2 transformation of values was applied to reduce skewness; ^‡^ No univariable logistic regression data available of this endpoint; ^+^ only *p*-values calculated with multivariable logistic regression are reported (including age, gender, diagnosis, and disease status as confounding factors). Bold indicates statistical significant values.

**Table 4 cancers-15-05443-t004:** Evaluation of the independent effects of EASIX score and ferritin levels on CRS, and of EASIX, ferritin, and CRP levels on ICANS.

		Parameters							
	Risk Group	EASIX	Ferritin	*n* *	Events, *n* *	CumInc, % *	HR	95% CI	*p*
CRS grade ≥ 2	High risk	High	Any level	17	8	47	0.96	0.43–2.12	0.92
	Intermediate risk	Low	High	41	13	32	0.87	0.47–1.60	0.64
	Low risk	Low	Low	21	9	43	1.00	-	-
		**Ferritin**	**EASIX/CRP**						
ICANS grade ≥ 2	High risk	High	Any Level	14	8	57	1.64	0.82–3.26	0.16
	Intermediate risk	Low	High EASIX	14	12	86	2.04	1.26–3.32	**<0.01**
			High CRP	14	5	36			
	Low risk	Low	Low EASIX and low CRP	29	11	38	1.00	-	-

* Only patients with available EASIX scores, and ferritin or CRP levels were eligible for this value. CumInc = cumulative incidence; CRP = C-reactive protein; EASIX = endothelial activation and stress index; HR = hazard ratio; CI = confidence interval; and *p* = *p*-value.

## Data Availability

The data presented in this study are available from the corresponding author upon reasonable request.

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
