# Peer review of "Population-Based External Validation of the EASIX Scores to Predict CAR T-Cell-Related Toxicities"

_cancers, 2023, doi:10.3390/cancers15225443_

Round 1

Reviewer 1 Report

Comments and Suggestions for Authors

Dr. de Boer et al present interesting results on their attempts to validate various iterations of the EASIX score in a population based cohort of patients with large B cell lymphoma treated with axi-cel in the Netherlands. This is a well written manuscript with a thoughtful analysis. I have a few comments or questions related to the manuscript. 

1) I would recommend including a little more about how EASIX score is calculated and which lab values are used for the calculation. I know you had references but I think may be helpful to have this information in the manuscript. 

2) In terms of patient characteristics, was the stage at diagnosis? What about the IPI score? Was the ECOG reported at time of lymphodepletion?

3) For LDH, would include upper limit of normal as this varies per institution. 

4) For patient characteristics, consider including median number of prior lines of therapy. For prior stem cell transplant, were these autos or allos?

5) Do you have information about number of patients who received steroids? Also, did any patients in this cohort receive steroid or tocilizumab prophylaxis?

6) What period was generally baseline for these patients? Was it at time of collection? What was the general time period between baseline and pre-lymphodepletion?

7) For the first sentence in section 3.3, may consider rewording that the EASIX pre-LD and at infusion were significantly associated with ICANS grade >2. 

8) When mentioning significant association of gender with toxicity, would state that its male gender in the text (can see it in supplement). 

9) Did any patients die from toxicity? 

Reviewer 2 Report

Comments and Suggestions for Authors

The authors of the manuscript “Population-based external validation of the EASIX scores to predict CAR T-cell related toxicities” described the use of the endothelial activation and stress index (EASIX) score and its derivatives, to assess the risk of Cytokine release syndrome (CRS) and immune effector cell-associated neurotoxicity syndrome (ICANS) in patients with relapsed/refractory large B-cell lymphoma treated with CAR T-cell therapy. 

The manuscript is well structured, the data shown are exhaustive and the figures are clear and contribute to a better understanding of the topic.

Minor criticisms:

1. the authors should include a pargraph describing the limitations of the study (e.g. the population of patients was treated  with only one of the currently available CAR-T cell therapies )

2. there are a minor typos to correct (e.g. line 104  “after ≥ 2 “)

Round 2

Reviewer 1 Report

Comments and Suggestions for Authors

Thank you for addressing my comments. I have no additional suggestions. 

Author Response

Dear reviewer 1,

I am pleased to hear that your comments have been adressed to your satisfaction. Thank you again for providing feedback. 

Best regards,

Janneke de Boer